# Development of a short-form Swedish version of the Montreal Cognitive Assessment (s-MoCA-SWE): protocol for a cross-sectional study

Tamar Abzhandadze [ID],[1,2] Erik Lundström,[3] Dongni Buvarp,[1] Marie Eriksson,[4] Terence J Quinn,[5] Katharina S Sunnerhagen [ID] [1,6]

► Prepublication history and supplemental material for this paper is available online. To view these files, please visit the journal online (http://dx.doi.org/10.1136/bmjopen-2021-049035).

For numbered affiliations see end of article.

**Correspondence to**
Dr Tamar Abzhandadze;
tamar.abzhandadze@gu.se

## ABSTRACT

**Introduction** Short forms of the Montreal Cognitive Assessment (MoCA) have allowed quick cognitive screening. However, none of the available short forms has been created or validated in a Swedish sample of patients with stroke.

The aim is to develop a short-form Swedish version of the MoCA (s-MoCA-SWE) in a sample of patients with acute and subacute stroke. The specific objectives are: (1) to identify a subgroup of MoCA items that have the potential to form the s-MoCA-SWE; (2) to determine the optimal cut-off value of s-MoCA-SWE for predicting cognitive impairment and (3) and to compare the psychometric properties of s-MoCA-SWE with those of previously developed MoCA short forms.

**Methods and analysis** This is a statistical analysis protocol for a cross-sectional study. The study sample will comprise patients from Väststroke, a local stroke registry from Gothenburg, Sweden and Efficacy oF Fluoxetine—a randomisEd Controlled Trial in Stroke (EFFECTS), a randomised controlled trial in Sweden. The s-MoCA-SWE will be developed by using exploratory factor analysis and the boosted regression tree algorithm. The cut-off value of s-MoCA-SWE for impaired cognition will be determined based on binary logistic regression analysis. The psychometric properties of s-MoCA-SWE will be compared with those of other MoCA short forms by using cross-tabulation and area under the receiving operating characteristic curve analyses.

**Ethics and dissemination** The Väststroke study has received ethical approval from the Regional Ethical Review Board in Gothenburg (346–16) and the Swedish Ethical Review Authority (amendment 2019–04299). The handling of data generated within the framework of quality registers does not require written informed consent from patients. The EFFECTS study has received ethical approval from the Stockholm Ethics Committee (2013/1265-31/2 on 30 September 2013). All participants provided written consent. Results will be published in an international, peer-reviewed journal, presented at conferences and communicated to clinical practitioners in local meetings and seminars.

## INTRODUCTION

Cognitive impairment, a common sequela of stroke, has been defined as a new cognitive

---

### Strengths and limitations of this study

► A short form of the Swedish version of the Montreal Cognitive Assessment will be created for the first time for use in patients with stroke.
► The study will use a large sample size from different acute stroke and stroke rehabilitation units across Sweden.
► The retrospective analysis of registry data is one limitation of the study.
► A high number of non-random missing Montreal Cognitive Assessment values is expected.

---

deficit that develops in the first 3 months after stroke and cannot be explained by other conditions or diseases.[1] Cognitive impairment is a consequence of complex interactions between age, education and injury size and location.[2] The prevalence of cognitive impairment in the first 3 weeks after stroke can vary between 55% and 59% depending on the time of assessment and the assessment instruments.[3 4] Previous studies have shown that cognitive impairment after stroke is associated with higher mortality and long-term disability.[5] Thus, early identification of cognitive impairments after stroke is important for planning individually tailored rehabilitation.

Comprehensive cognitive assessments in acute stroke units can be time and resource demanding for patients as well as for healthcare professionals. Therefore, the use of short screening tools for cognitive impairment is recommended.[6 7] The Montreal Cognitive Assessment (MoCA) is frequently used to screen cognitive function.[7] The MoCA targets cognitive domains such as visuospatial ability, executive function, attention, concentration, working memory, language, short-term memory and orientation.[8] It has a good sensitivity for detecting poststroke cognitive impairment.[9 10] To make cognitive screenings

time effective, several short forms of the MoCA have been developed and tested in recent years, including a telephone version of the MoCA (T-MoCA), a short version of the MoCA that has been designed by the National Institute of Neurological Disorders and Stroke and the Canadian Stroke Network (NINDS-CSN, 5 min protocol) and the short-form MoCA (SF-MoCA).[11–19] Different cutoffs for impaired cognition have been reported for the MoCA as well as its short forms.[11 12 15 17–20] None of the available short forms have been developed or validated in a Swedish sample of patients with stroke.

Performance on cognitive tests after stroke can depend on premorbid cognitive reserves, which are associated with education, occupation and socioeconomic situation.[21 22] Items of the MoCA can function differently in various cultures; therefore, cultural validation of the MoCA has been recommended for generalisability.[23] Theoretically, this recommendation is accurate even when it comes to short forms of the MoCA. In order to have a short form of the MoCA used in a broader context as a time-effective screening instrument with good sensitivity for detecting cognitive impairment, it needs to be validated culturally within the target population.

### Study aims

The aim of this study is to develop a short-form Swedish version of the MoCA (s-MoCA-SWE) using data from Swedish stroke cohorts. The specific objectives are: (1) to identify a subgroup of MoCA items that have the potential to form the s-MoCA-SWE; (2) to determine the optimal cut-off value of s-MoCA-SWE for predicting cognitive impairment and (3) to compare the psychometric properties of s-MoCA-SWE with those of T-MoCA, NINDS-CSN and SF-MoCA.

### METHODS AND ANALYSIS
### Study design

This is a cross-sectional, exploratory study. Two datasets will be used: Väststroke, a local stroke registry from Gothenburg, Sweden and Efficacy oF Fluoxetine—a randomisEd Controlled Trial in Stroke (EFFECTS), a randomised controlled trial.[24]

The Väststroke register comprises data from three stroke units in Sahlgrenska University Hospital. The data in Väststroke were gathered within the scope of the Physical Activity Pre-Stroke In GOThenburg (PAPSIGOT) project.[25] The Väststroke registry was linked to the Riksstroke, a national quality register for stroke care in Sweden, via each patient's unique personal identification number.[26] The statisticians at Riksstroke merged the data. The Väststroke data from 1 November 2014 to 30 June 2019 were retrieved.

The EFFECTS was a randomised, double-blind, placebo-controlled clinical trial conducted in 35 stroke or rehabilitation units in Sweden.[24] The inclusion period was from 20 October 2014 to 28 June 2019.[24] Briefly, patients aged ≥18 years and diagnosed with ischaemic or haemorrhagic stroke confirmed by brain imaging were included. Detailed information about EFFECTS can be found elsewhere.[24]

### Study sample

The Väststroke dataset comprises data on 6493 patients; the EFFECTS dataset comprises data on 1500 patients (online supplemental figure). The two datasets will be aligned.

The inclusion criteria are as follows:
► Age ≥18 years at stroke onset.
► A diagnosis of stroke according to the International Classification of Diseases-10: intracerebral haemorrhage (I61), cerebral infarction (I63) and stroke, not specified as haemorrhage or infarction (I64).
► Patients with a previous stroke and/or transient ischaemic attack (TIA).
► A complete MoCA (0–30 p).
The exclusion criterion was as follows:
► Duplicate registration on the Väststroke and EFFECTS datasets (duplicates will be removed from the Väststroke dataset).

### Study procedure

Väststroke data were registered by healthcare professionals working in stroke units. The MoCA was administered by occupational therapists. Occupational therapists at stroke units have regular workshops and peer reviews regarding the administration and interpretation of the MoCA. In the EFFECTS, the patients were enrolled in the stroke and rehabilitation units between 2 and 15 days after stroke onset. Cognitive screening was performed by study personnel such as nurses or physicians at the local centre, without any formal training of the instrument; sometimes, the assessment by the local occupational therapist was used. Detailed information about the EFFECTS study is available elsewhere.[27] Cognitive screening was performed during the acute and early subacute phases of stroke.

### Study variables

Only the variables available in both the Väststroke and EFFECTS datasets will be used. Detailed information on all variables that will be included in the study, their categories, and coding for the merged datasets are presented in online supplemental table 1.

The MoCA is a valid and reliable instrument for cognitive screening in the acute and subacute phases of stroke.[8–10] The total MoCA score ranges between 0 and 30 points (p); 1 p for a maximum of 12 years of education can be added to the total score. A score of ≤25 p indicates cognitive impairment.[8] MoCA items were registered in the datasets as follows:
► Orientation (range, 0–6 p).
► Delayed recall (range, 0–5 p).
► Visuospatial/executive functions (range, 0–5 p): trial (1 p)+cube (1 p)+clock (3 p).
► Naming of three animals (range, 0–3 p).

- ► Digit span, two tasks (range, 0–2 p).
- ► Repetition of two sentences (range, 0–2 p).
- ► Categories, two tasks (range, 0–2 p).
- ► Serial 7 (range, 0–3 p).
- ► Fluency (range, 0–1 p).
- ► Tap on A (range, 0–1 p).

Stroke severity at onset was assessed using the National Institutes of Health Stroke Scale (NIHSS).[28] The NIHSS comprises 15 items with varying score ranges per item. The total NIHSS score is 0–42 p, with a higher score indicating a more severe stroke.[28]

Reperfusion treatment will be categorised as both thrombolysis and thrombectomy, only thrombolysis or only thrombectomy.

Information about a previous stroke and/or TIA was recorded (yes, no or unknown). The Väststroke register comprises patients with a first-ever stroke, however, patients with previous TIA were included. In the EFFECTS study, patients could have a prior stroke and/or TIA, and these two conditions could not be distinguished.

### Aggregated variables

The T-MoCA is a short version of the MoCA that is based on the following items: delayed recall, digit span, list letters, serial 7, sentence repetition, fluency, abstraction and orientation.[14] The total T-MoCA score is 22 p. The reported threshold for impaired cognition is ≤18 p.[14]

The NINDS-CSN is based on items such as delayed recall, fluency, and orientation.[15 17–19] The total NINDS-CSN score is 12 p. The suggested thresholds for impaired cognition are ≤9 p,[13 15] ≤10 p[16] and ≤6 p.[17 29]

The SF-MoCA is based on items such as delayed recall, serial 7 and orientation.[11 12] The total SF-MoCA score is 14 p, and a cut-off of ≤8 p has been suggested as an indicator for impaired cognition.[11 12]

### Statistics

Data will be presented as the mean and SD, median and IQR, minimum–maximum (min–max), number (n) and proportion (%) (online supplemental tables 2 and 3). The OR and 95% CI will be presented when appropriate. The significance level for the statistical tests will be set at $\alpha=5\%$. All statistical tests will be two tailed.

### Objective 1

Two different approaches will be used to identify the MoCA items that have the potential to form the s-MoCA-SWE: exploratory factor analysis[30 31] and boosted regression tree.[32]

Exploratory factor analysis is a dimension-reduction method with good sensitivity for identifying the latent connections between variables. Exploratory factor analysis will be performed in several steps. First, the correlations between the MoCA items will be calculated. Second, the results of Bartlett's and Kaiser-Meyer-Olkin tests will be evaluated; values lower than 0.05 and greater than 0.7, respectively, are desirable. Third, the number of factors will be determined based on the scree plot and parallel analysis scree plot of eigenvalues. A factor-extraction method will be chosen, and a decision about rotation will be made based on the results of the previous steps. MoCA items with loading >0.6 will be chosen as having the potential to enter the s-MoCA-SWE.[31]

Boosted regression tree is a supervised machine-learning algorithm. In the boosted regression tree algorithm, modelling trees are grown sequentially (eg, each tree is grown based on the information from the previously grown trees and each tree is fit to the residuals from the previous tree). For this analysis, the dataset will be split into training (80%) and test (20%) datasets. The s-MoCA-SWE boosted regression tree will be developed based on the training dataset, and tuning parameters such as the number of trees (B), shrinkage parameter ($\lambda$) and interaction depths (d) will be chosen based on 10-fold cross-validation and bootstrapping. The MoCA items with the potential to build an s-MoCA-SWE will be chosen based on the boosted regression tree model with adjusted tuning parameters and variable coefficients >5.0. The s-MoCA-SWE developed based on the training dataset will be further tested on the test dataset.

The decision between the set of MoCA items derived from exploratory factor analysis and that derived from the boosted regression tree algorithm will be made by applying binary logistic regression analysis. The sets will be entered as independent variables and tested using different regression models. The dependent variable will be dichotomised MoCA (≤25 p for impaired cognition).[8 33] The s-MoCA-SWE version with the highest sensitivity and classification accuracy for cognitive impairment will be analysed in objective 2.

### Objective 2

To find the optimal cut-off value of s-MoCA-SWE for classifying patients with and without cognitive impairment, binary logistic regression analysis will be performed. The MoCA cut-off of ≤25 p for impaired cognition will be the reference output,[8 33] and the s-MoCA-SWE full score will be the independent variable. The cut-off value of s-MoCA-SWE will be chosen based on its good sensitivity for identifying patients with cognitive impairment.

### Objective 3

Four cross-tables will be evaluated. The reference variable will be dichotomised MoCA (≤25 p for impaired cognition).[8 33] The index variables will be dichotomised s-MoCA-SWE (according to the results from Objective 2), T-MoCA (≤18 p for impaired cognition),[14] NINDS-CSN (≤9 p for impaired cognition)[13 15] and SF-MoCA (≤8 p for impaired cognition).[11 12] The psychometric measurements of sensitivity, specificity, positive predictive value, negative predictive value and Youden's index with 95% CI will be evaluated.

Receiving operating characteristic curve analysis will be performed. The full scores of s-MoCA-SWE, T-MoCA, NINDS-CSN and SF-MoCA will be entered as test variables, and the dichotomised MoCA (≤25 p for impaired

cognition) will be entered as the state variable.[8 33] The areas under the curve will be interpreted as follows: 0.7–0.9, moderate accuracy and 0.5–0.7, low accuracy.[34]

### Subgroup analyses

The properties of the s-MoCA-SWE will be evaluated further. Subgroup analysis will be performed by studying the differences between patients with and without a previous stroke/TIA, with the hypothesis that patients with a previous stroke are more likely to have impaired cognition; by stroke severity, with the hypothesis that patients with more severe stroke are more likely to have more impaired cognition; and by age, with the hypothesis that older patients are more likely to have impaired cognition.

### Missing data

For descriptive statistics, data on all included patients will be used. The proportion of missing values will be provided for each baseline variable.

### Statistical software

Data will be analysed using SPSS Statistics for Windows V.27.0 (released on 2018; IBM) and R V.4.0.2 (R Core Team. R: A language and environment for statistical computing. R Foundation for Statistical Computing, Vienna, Austria.).

### Patient and public involvement statement

The patients and the public were not involved in the design of the protocol.

Cognition has been identified by patients, their caregivers, and healthcare professionals as one of the most important research areas in stroke.[35] Furthermore, many patients have reported unmet needs regarding problems with cognitive functions, such as memory, concentration and speaking/reading.[36] Thus, early identification of cognitive impairment is important.

Patients with stroke can find cognitive assessments fairly burdensome in acute stroke and would prefer short tests if possible. By developing s-MoCA-SWE, we hope to have a simpler and more efficient cognitive screening tool that can be used during acute and subacute stroke. Theoretically, this would lead to a higher proportion of patients who receive the recommended cognitive screenings.

### ETHICS AND DISSEMINATION

Väststroke: The study obtained ethics approval from the Regional Ethical Review Board in Gothenburg (346–16) and the Swedish Ethical Review Authority (amendment 2019–04299). According to the Swedish Data Protection Authority, the handling of data generated within the framework of quality registers represents an exception to the general rule of written informed consent being required from patients. Furthermore, the Personal Data Act (Swedish law #1998:204, issued 29 April 1998) allows data from medical charts to be collected for clinical purposes and quality control without written informed consent.

The EFFECTS study has received ethical approval from the Stockholm Ethics Committee (2013/1265-31/2 on 30 September 2013). All participants of the EFFECTS study provided written consent for inclusion in the randomised controlled trial. They also provided written and oral consent to merge their data with other registry data.

In order to disseminate the study findings to a wide audience, results will be published in international, peer-reviewed journals, presented at local and international conferences, communicated to clinical practitioners in local meetings and seminars and shared via social media channels.[37] Furthermore, a short report will be published in Swedish journals that target professionals who perform clinical cognitive screenings.

**Author affiliations**
[1]Institute of Neuroscience and Physiology, The Sahlgrenska Academy, University of Gothenburg, Gothenburg, Sweden
[2]Department of Occupational Therapy and Physiotherapy, Sahlgrenska University Hospital, Gothenburg, Sweden
[3]Department of Neuroscience, Neurology, Uppsala University, Akademiska Sjukhuset, Uppsala, Sweden
[4]Department of Statistics, USBE, Umeå University, Umeå, Sweden
[5]Institute of Cardiovascular and Medical Sciences, University of Glasgow, Glasgow, UK
[6]Neurocare, Sahlgrenska University Hospital, Gothenburg, Sweden

**Contributors** TA: conceptualisation of the study, statistics, drafting of the manuscript; EL: chief investigator of EFFECTS, providing of EFFECTS's data, critical revision of the manuscript; DB: statistical consultations, critical revision of the manuscript; ME: statistical consultations, critical revision of the manuscript; TJQ: statistical consultations, conceptualisation of the study, critical revision of the manuscript; KSS: acquisition of Väststroke data, conceptualisation of the study, critical revision of the manuscript.All authors read and approved final version of the manuscript.

**Funding** This work was supported by the IRIS foundation, Swedish Research Council (VR2017-00946), Swedish Heart and Lung Foundation, Swedish Brain foundation, Promobilia, Swedish state under an agreement between the Swedish government and the county councils, ALF agreement (ALFGBG-718711, ALFGBG-877961), Swedish National Stroke Association, Local Research and Development Board for Gothenburg and Södra Bohuslän, Greta and Einar Asker's foundation, Rune and Ulla Almöv's foundation for neurological research, Hjalmar Svensson's Research foundation, P-O Ahl & J-B Wennerström's foundation; Herbert & Karin Jacobssons foundation, Sahlgrenska University Hospital foundations, Wilhelm and Martina Lundgren's foundation, and Inger Bendix's foundation for medical research.

**Competing interests** None declared.

**Patient consent for publication** Not required.

**Provenance and peer review** Not commissioned; externally peer reviewed.

ORCID iDs
Tamar Abzhandadze http://orcid.org/0000-0002-0069-6875
Katharina S Sunnerhagen http://orcid.org/0000-0002-5940-4400

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
