## [Reviewer comments · BMJ Open]

ARTICLE DETAILS

TITLE (PROVISIONAL)	Development of a short-form Swedish version of the Montreal Cognitive Assessment (s-MoCA-SWE): Protocol for a cross-sectional study
AUTHORS	Abzhandadze, Tamar; Lundström, Erik; Buvarp, Dongni; Eriksson, Marie; Quinn, Terry; Sunnerhagen, Katharina

VERSION 1 – REVIEW

REVIEWER	Cullum, Munro University of Texas Southwestern Medical Center at Dallas, Department of Psychiatry
REVIEW RETURNED	04-Feb-2021

GENERAL COMMENTS	The authors describe a series of planned analyses to develop a Swedish version of the MoCA for use in patients following stroke, to determine an optimal cutoff score for predicting (actually identifying) cognitive impairment, and to compare the psychometric properties of the short form with other MoCA short forms. The study design is sound, and the authors clearly have access to the databases needed to achieve their aims. What is puzzling to this reviewer is that there are no data, analyses, or results, and this paper really just describes what the authors plan to do. As such, it is quite unclear what such a paper would add to the literature, particularly since other MoCA short forms have been published that describe their development process. This study would use existing methods and statistics to create a Swedish short form MoCA, which may be useful, with without any data, analyses, results, or discussion, it is difficult to understand what the paper in its current form offers readers. In fact, in reading the paper, I was wondering if the results and discussion sections were somehow omitted from the submission, as it appears incomplete. As such, as a purely proposed "methods" paper, I am not sure what the paper would add to the literature or how this would be of interest to readers of this journal.
---

REVIEWER	Dupuis, Kate Sheridan College, Sheridan Centre for Elder Research
REVIEW RETURNED	23-Feb-2021

GENERAL COMMENTS	Abstract: -Please specify whether you are interested in evaluating potential language differences with an existing MoCA stroke
---

	version? Is there already a stroke tool or are you a) creating a new, shorter tool that is specific to b) stroke population and c) Swedish language? Please clarify exactly what is novel for this work -are you recruiting only from the control arm of the EFFECTS study? Could there be potential differences in how pts on Fluoxetine react on the s-MoCA-SWE? p. 6 Strengths and Limitations -It appears to me as though you are trying to accomplish two goals here 1) entirely new Swedish version and 2) norm on a new population. Can you please clarify and ensure that is reflected in the abstract? Is there a MoCA-Stroke? Or are you creating one? -p. 7 line 15- cognitive impairment associated... in general or specifically after a stroke? -p. 7 lines 15-16- do you mean early intervention is important? Study design, -p.8, no mention of Fluoxetine? -Please describe whether pts in the Väststroke had also received SSRI therapy post-stroke? -How can you merge datasets from two such different study designs? -a complete MoCA- is this the traditional /30 MoCA, translated to Swedish? -are all participants first language Swedish? Do any of them speak different languages? -Study procedure: how did you control for potential errors in test administration due to the fact that OTs in one sample were trained to administer the MoCA yet nurses/MDs in the other sample were not? -p.10 you say that the V sample consisted of pts. with first-ever strokes; these pts could present very differently than those with an extensive history of strokes. How can you compare these two groups? How did you control for this?
--	---

REVIEWER	Diaz-Orueta, Unai Maynooth University
REVIEW RETURNED	28-Feb-2021

GENERAL COMMENTS	It is a very well written protocol, with all the steps and procedures clearly defined. Great job!
---

VERSION 1 – AUTHOR RESPONSE

Questions	Answers
Reviewer: 1, Dr. Munro Cullum, University of Texas Southwestern Medical Center at Dallas	
The authors describe a series of planned analyses to develop a Swedish version of the MoCA for use in patients following stroke, to determine an	Thank you for the feedback. Our manuscript is a statistical analysis protocol for a cross-sectional study. According to instructions “Protocol manuscripts should report planned or ongoing research studies”. However, according to the guidelines

optimal cutoff score for predicting (actually identifying) cognitive impairment, and to compare the psychometric properties of the short form with other MoCA short forms. The study design is sound, and the authors clearly have access to the databases needed to achieve their aims. What is puzzling to this reviewer is that there are no data, analyses, or results, and this paper really just describes what the authors plan to do. As such, it is quite unclear what such a paper would add to the literature, particularly since other MoCA short forms have been published that describe their development process. This study would use existing methods and statistics to create a Swedish short form MoCA, which may be useful, with without any data, analyses, results, or discussion, it is difficult to understand what the paper in its current form offers readers. In fact, in reading the paper, I was wondering if the results and discussion sections were somehow omitted from the submission, as it appears incomplete. As such, as a purely proposed "methods" paper, I am not sure what the paper would add to the literature or how this would be of interest to readers of this journal.	we can not start analyses and report the results prior to publication of the protocol. The manuscript is written according to the BMJ open guidelines stated on the webpage: https://bmjopen.bmj.com/pages/authors/ under the heading "Protocol". The manuscript should have following parts: Title, abstract, an Article Summary, placed after the abstract, consisting of the heading 'Strengths and limitations of this study', introduction, methods and analysis, ethics and dissemination, full references, authors' contributions, funding statement, and competing interests' statement.
--	---

Reviewer: 2, Dr. Kate Dupuis, Sheridan College

Abstract: -Please specify whether you are interested in evaluating potential language differences with an existing MoCA stroke version? Is there already a stroke tool or are you a) creating a new, shorter tool that is specific to b) stroke population and c) Swedish language? Please clarify exactly what is novel for this work -are you recruiting only from the control are of the EFFECTS study? Could there be potential differences in how pts on Fluoxetine react on the s-MoCA-SWE?	Thank you for asking the question. The MoCA is already translated into Swedish and is commonly used as a first screening instrument in patients with stroke. In this project we want to create a Swedish short MoCA for patients who have had a stroke and find the cut-of values for impaired cognition. However, we also want to validate short MoCAs that were previously developed in other countries. We can not speak about effects of Fluoxetine at this stage. The screening with MoCA was done prior to randomisation in EFFECTS. Regarding the Vaststroke, some of the patients could have been on SSRI pre-stroke. However, this is true of any stroke population and should not confound results. Across all the recent SSRI stroke trials a cognitive enhancing effect of SSRI has not been demonstrated. We have made a clarification in the abstract, p. 2, introduction: - However, none of the available short forms has been created or validated in a Swedish sample of patients with stroke.
p. 6 Strengths and Limitations. It appears to me as though you are trying to accomplish two goals here 1) entirely new	Thank you for asking the clarification. We aim to develop a short-form of MoCA. As no previous Swedish short-MoCAs exist this will be entirely new Swedish version. The cut-off values for

Swedish version and 2) norm on a new population. Can you please clarify and ensure that is reflect in the abstract? Is there a MoCA-Stroke? Or are you creating one?	impaired cognition will be calculated for Swedish sample with stroke. We have revised the first sentence of the Strengths and Limitation section, p. 4, line 2:  - A short form of the Swedish version of the Montreal Cognitive Assessment will be created for the first time for use in patients with stroke.
p. 7 line 15- cognitive impairment associated... in general or specifically after a stroke?	Thank you for asking the clarification. We have added the text. Please read: introduction, page 5, line 6:  - Previous studies have shown that cognitive impairment after stroke is associated with higher mortality and long-term disability
p. 7 lines 15-16- do you mean early intervention is important?	We believe early identification of stroke-related cognitive impairments are important to allow early and targeted rehabilitation. We have revised the sentence. Please read: introduction, page 5, lines 7 and 8:  - Thus, early identification of cognitive impairments after stroke is important for planning individually tailored rehabilitation
Study design p.8, no mention of Fluoxetine?	The screening with MoCA was done prior to randomisation in EFFECTS. This is the major reason why we do not mention Fluoxetine in the study design.
Please describe whether pts in the Väststroke had also received SSRI therapy post-stroke?	Thank you for asking the question. The screening with MoCA was done prior to randomisation in EFFECTS. Regarding the Väststroke, some of the patients could have been on SSRI pre-stroke. However, this is true of any stroke population and should not confound results. Across all the recent SSRI stroke trials a cognitive enhancing effect of SSRI has not been demonstrated.
How can you merge datasets from two such different study designs?	Thank you for asking the question. The data files will be aligned with each other by using the same variables in both datasets. Furthermore, although the studies are of different design, we do believe this is methodological sound. Please find additional information, p.6, study sample, line 2.  - The two datasets will be aligned.
A complete MoCA- is this the traditional /30 MoCA, translated to Swedish?	Yes, complete MoCA is a traditional 30 p MoCA translated to Swedish. We made a clarification, please read: Study sample, p. 7, line 5:  - a complete MoCA (0 – 30 p).
Are all participants first language Swedish? Do any of them speak different languages?	This is a very good question. Some participants did not have Swedish as a first language. However, if included in the studies, they would have sufficient proficiency in Swedish to allow for meaningful testing with the MoCA screening tool.
Study procedure: how did you control for potential errors in test administration due to the fact that OTs in one sample were trained to administer the MoCA yet nurses/MDs in the other sample were not?	At the inception of both studies the official training in MoCA created by the copyright holders was not mandated or widely used. Assessors in both studies have training and experience in administration of the MoCA to local standards.
P.10 you say that the V sample consisted of pts. with first-ever strokes; these pts could present very differently than those with an extensive history of strokes. How can you compare these two groups? How did you control for this?	Thank you for asking the question. In the datasets there is one variable containing information if patients had previous stroke/TIA or not. We will use this variable for testing the null hypothesis that patients with previous stroke/TIA have the same risk of impaired cognition compared with those with first-time stroke.

	The text is added for more clarification. Please read: Study variables, page 8, lines 14-15.  - Information about a previous stroke and/or TIA was recorded (yes, no, or unknown). The Väststroke register comprises patients with a first-ever stroke, however patients with previous TIA were included. In the EFFECTS study, patients could have a prior stroke and/or TIA, and these two conditions could not be distinguished
Reviewer: 3, Dr. Unai Diaz-Orueta, Maynooth University	
It is a very well written protocol, with all the steps and procedures clearly defined. Great job!	Thank you very much.

VERSION 2 – REVIEW

REVIEWER	Dupuis, Kate Sheridan College, Sheridan Centre for Elder Research
REVIEW RETURNED	29-Mar-2021
GENERAL COMMENTS	Thank you for your clear and concise responses to the initial reviews.